# Nutrient Patchiness, Phytoplankton Surge-Uptake, and Turbulent History: A Theoretical Approach and Its Experimental Validation

**Mathilde Schapira [1,*,†] and Laurent Seuront [2,3,4,*,†]**

[1] Ifremer, LITTORAL, F-44311 Nantes, France
[2] CNRS, Univ. Lille, Univ. Littoral Côte d'Opale, UMR 8187, LOG, Laboratoire d'Océanologie et de Géosciences, F-62930 Wimereux, France
[3] Department of Marine Resource and Energy, Tokyo University of Marine Science and Technology, 4-5-7 Konan, Minato-ku, Tokyo 108-8477, Japan
[4] Department of Zoology and Entomology, Rhodes University, Grahamstown 6140, South Africa
[*] Correspondence: mathilde.schapira@ifremer.fr (M.S.); laurent.seuront@cnrs.fr (L.S.)
[†] These authors contributed equally to this work.

**Abstract:** Despite ample evidence of micro- and small-scale (i.e., millimeter- to meter-scale) phytoplankton and zooplankton patchiness in the ocean, direct observations of nutrient distributions and the ecological importance of this phenomenon are still relatively scarce. In this context, we first describe a simple procedure to continuously sample nutrients in surface waters, and subsequently provide evidence of the existence of microscale distribution of ammonium in the ocean. We further show that ammonium is never homogeneously distributed, even under very high conditions of turbulence. Instead, turbulence intensity appears to control nutrient patchiness, with a more homogeneous or a more heterogeneous distribution observed under high and low turbulence intensities, respectively, under the same concentration in nutrient. Based on a modelling procedure taking into account the stochastic properties of intermittent nutrient distributions and observations carried out on natural phytoplankton communities, we introduce and verify the hypothesis that under nutrient limitation, the "*turbulent history*" of phytoplankton cells, i.e., the turbulent conditions they experienced in their natural environments, conditions their efficiency to uptake ephemeral inorganic nitrogen patches of different concentrations. Specifically, phytoplankton cells exposed to high turbulence intensities (i.e., more homogeneous nutrient distribution) were more efficient to uptake high concentration nitrogen pulses (2 µM). In contrast, under low turbulence conditions (i.e., more heterogeneous nutrient distribution), uptake rates were higher for low concentration nitrogen pulses (0.5 µM). These results suggest that under nutrient limitation, natural phytoplankton populations respond to high turbulence intensities through a decrease in affinity for nutrients and an increase in their transport rate, and vice versa.

**Keywords:** nutrient patchiness; turbulence; phytoplankton; surge uptake; nutrient depletion; turbulent history

## 1. Introduction

Investigations of micro- to small-scale (typically millimeter- to meter-scale) distributions of viruses, bacteria, phytoplankton, and zooplankton populations revealed their patchy character, see e.g., [1–22], in particular in relation with turbulence [23–25]. Despite early attempts to quantify nutrient patchiness in the ocean [26,27], the introduction of a simple procedure to continuously sample nutrients from surface waters nearly two decades ago [28], and the plethora of work devoted to the assessment of

the implication of microscale nutrient patchiness for aquatic microbial communities [29–35], direct observations of micro-scale nutrient distributions are still lacking. Information on the qualitative and quantitative nature of micro-scale nutrient distribution is nevertheless critically needed to bridge the gap between bacteria and phytoplankton and higher trophic levels to improve our general understanding of structures and functions in marine systems.

The existence of nutrient microzones has long been hypothesized to result in the development of phytoplankton adaptive strategies for nutrient uptake [36–38]. However, little is still known about the potential effect of the interplay between nutrient patchiness and phytoplankton uptake in natural waters. The uptake of nutrient by phytoplankton cells is typically described using the Monod equation, i.e., a strict equivalent to the Michaelis-Menten equation. These models both fundamentally hypothesized steady state conditions, i.e., a homogenous distribution of the limiting nutrient in time. Under non-steady state conditions, such as nutrient patchiness, these equations cannot represent correctly nutrient removal by phytoplankton cells and to the best of our knowledge there is no experimentally validated model of nutrient uptake under fluctuating nutrient conditions [39]. Nutrient patchiness may have a negative impact on nutrient uptake rates as uptake is typically less efficient at higher nutrient concentration than at low ones [36]. This hypothesis holds true under the general assumption that the parameters of the Michaelis-Menten kinetics remain constant irrespective of ambient nutrient concentration [39]. It becomes, however, fundamentally unrealistic given the well-known abilities of nutritionally limited phytoplankton cells to enhance their uptake of nutrients in the presence of ephemeral point source [40–42].

In this context, the first aim of this work is to briefly rehearse the description of a simple technique allowing the continuous sampling of nutrient from surface waters [28], to critically assess its potential limitations, and to illustrate its validity to characterize nutrient patchiness in the specific framework of sampling experiments conducted in the eastern English Channel. Our second aim was thus to introduce a modelling procedure that might account for the observed surge-uptake of nutrient based (i) on the detailed stochastic properties of intermittent nutrient distributions, and (ii) on a simple adaptive representation of phytoplankton surge-uptake for nutrients. Finally, we validate our mechanistic hypotheses through a specifically designed field experiment devoted to assess the surge uptake rates of natural phytoplankton communities under ammonium limitations when exposed to ammonium pulses of low and high concentrations.

## 2. Assessing Nutrient Patchiness in the Ocean

### 2.1. High-Frequency Nutrient Sampling

In order to continuously investigate the small-scale distribution of ammonium, a series of three sampling experiments have been conducted adrift in the coastal waters of the eastern English Channel in the summers of 1996, 1997, and 1998. Water was continuously taken from a depth of 0.25 m through a seawater intake mounted on a suspended hose located 1 m away from the hull of the vessel, and directly processed in a Technicon Autoanalyzer II [43] by means of a rail wheel pump connected to 1.5-mm diameter plastic tubing with an approximate output of 0.80 mL min$^{-1}$. The temporal resolution (i.e., 3 s) was chosen as the minimum time interval allowed by the Technicon Autoanalyzer II between two ammonium quantity determinations. Despite early suggestions of its feasibility [43], the approach described in the present work has, to our knowledge, only been used at a lower temporal resolution (1 min) and a smoothing of the output signal associated with the dimension of the pumping apparatus [26], and for the determination of nitrites micro-scale distribution [28]. Data were directly recorded on a PC by means of a data logger system interfaced with the Technicon Autoanalyzer II. Between each time series, the whole plastic tubing was rinsed with HCl 10%, Milli-Q water and the Technicon Autoanalyzer II was calibrated using a standardized nitrogen solution. We recorded 11, 15, and 8 time series of ammonium concentrations of approximately 1-h duration at a sampling frequency of 0.33 Hz in 1996, 1997, and 1998, respectively. Sampled time series show the very intermittent

character of ammonium distribution (i.e., a distribution characterized by a few dense patches and a wide range of low-density patches; [28]), whatever the year and the hydrodynamic conditions (Figure 1).

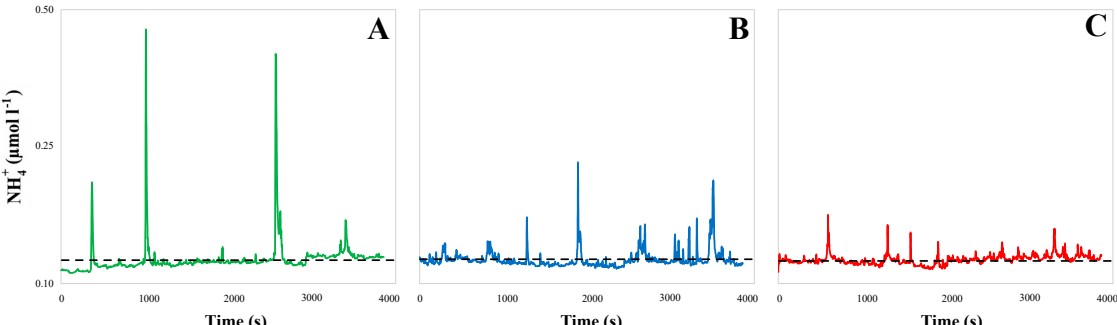

**Figure 1.** Samples of high-resolution (0.33 Hz) ammonium concentration time series recorded in the eastern English Channel in 1998 for increasing values of the tidal current speed $v$ (m s$^{-1}$): $v \sim 0.2$ m s$^{-1}$ ($\varepsilon \sim 6 \times 10^{-4}$ m$^2$ s$^{-3}$, (**A**)), $v \sim 0.5$ m s$^{-1}$ (i.e., $\varepsilon \sim 7.5 \times 10^{-5}$ m$^2$ s$^{-3}$, (**B**)), and $v \sim 1$ m s$^{-1}$ ($\varepsilon \sim 4.8 \times 10^{-6}$ m$^2$ s$^{-3}$, (**C**)). The ammonium distributions are patchier when the hydrodynamic conditions (i.e., the tidal current speed $v$) are weaker. The dissipation rate of turbulent energy $\varepsilon$ (m$^2$ s$^{-3}$) were estimated from the velocity of the tidal flow $v$ as $\varepsilon = 0.006$ ($v^3/z$), where $z$ is the depth of the water column ($z \sim 10$ m); see Equation (15) below. The black dashed lines are the average of each distribution, and illustrate the extreme similarity in the bulk concentration of ammonium.

To assess the presence of a potential link between ammonium and phytoplankton biomass that may bias further analysis and interpretation of the distribution of ammonium, the 11, 15, and 8 times series of ammonium concentrations sampled in 1996, 1997, and 1998 were consistently sampled simultaneously to in vivo fluorescence (a proxy of phytoplankton biomass) using a Sea Tech fluorometer, and both temperature and salinity using a Sea-Bird Sealogger CTD. Though the differences in the level of small-scale patchiness observed in nutrients, in vivo fluorescence and purely passive tracers such as temperature and salinity have been discussed at length elsewhere [4,11], we briefly rehearse (i) that we never found any significant correlation ($p > 0.05$) between ammonium times series, and neither in vivo fluorescence, temperature, nor salinity, and (ii) that the stochastic properties of temperature and salinity were consistent with the signature of purely passive tracers advected by turbulent velocity fluctuations, in contrast to both in vivo fluorescence and nutrients that exhibit very distinct levels of patchiness, i.e., more and less patchy under low and high turbulent conditions, respectively [11,28]. These observations warrant that, though ammonium is, *a priori,* a far less conservative nutrient than, e.g., nitrate and nitrite [28], it can then be considered as independent from the phytoplankton and the turbulent fields at the temporal scales considered in the present work.

*2.2. Potential Sources of Aliasing and Validation*

In order to get a reference framework to validate our continuous use of the Technicon Autoanalyzer II, we plotted the mean of each time series together with ammonium concentrations estimated from triplicated sub-surface Niskin bottle samples taken within 10 cm from the continuous seawater intake at the beginning, the middle and the end of each time series record (Figure 2). Both the very good agreement observed between the ammonium concentration obtained from our novel continuous sampling and a standard discrete sampling scheme and the lack of interannual variability in the observed agreement suggest that the occurrence of the observed high-density patches (cf. Figure 1) cannot be attributed to any kind of aliasing. Several potential sources of aliasing that can be proposed for the occurrence of these patches are nevertheless discussed hereafter.

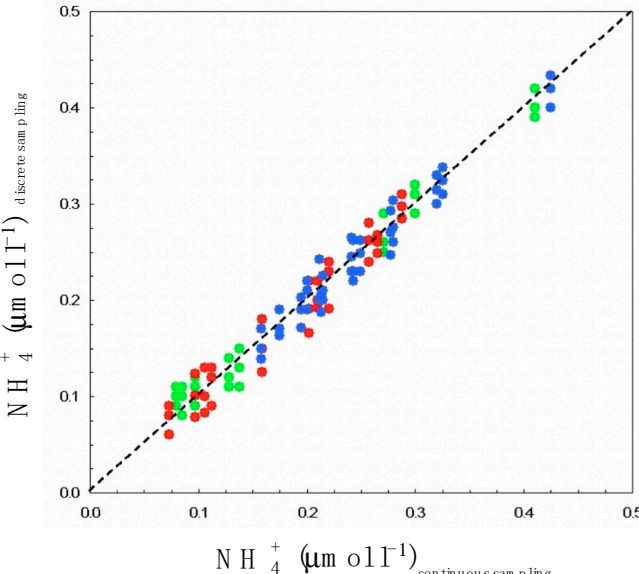

**Figure 2.** Mean values of ammonium times series recorded in 1996, 1997, and 1998, plotted against the ammonium concentrations simultaneously estimated from sub-surface Niskin bottle samples. The resulting highly significant linear regression ($p < 0.01$) demonstrates the validity of our high-resolution sampling procedure. The different colors correspond to sampling under different flow speed $v$ conditions: red ($v \sim 1$ m s$^{-1}$; i.e., $\varepsilon \sim 6 \times 10^{-4}$ m$^2$ s$^{-3}$), blue ($v \sim 0.5$ m s$^{-1}$, i.e., $\varepsilon \sim 7.5 \times 10^{-5}$ m$^2$ s$^{-3}$), and green ($v \sim 0.2$ m s$^{-1}$, i.e., $\varepsilon \sim 4.8 \times 10^{-6}$ m$^2$ s$^{-3}$). The dissipation rate of turbulent energy $\varepsilon$ (m$^2$ s$^{-3}$) were estimated from the velocity of the tidal flow $v$ as $\varepsilon = \phi v^3/z$, where $z$ is the depth of the water column ($z \sim 10$ m) and $\phi$ is a constant, $\phi = 0.006$; see Equation (15) below.

As briefly investigated by Seuront et al. [28], the four main sources of aliasing that can be identified in our sampling procedure are discussed hereafter.

### 2.2.1. The Motion of the Ship

A contamination by the motion of the ship implies that each point of a recorded dataset may potentially have been sampled at a different depth. Such a bias would typically be related to the characteristic frequency of the waves, and consequently would have led to peaks at specific frequencies when plotting the ammonium time series in Fourier space [44], which is obviously not the case (Figure 3). Note, however, that even if the observed ammonium patches could have been related to the rolling of the ship, their ecological relevance would be equivalent, providing evidence for vertical small-scale patchiness instead of/and a horizontal one.

### 2.2.2. The Characteristics of the Sample Processing Chain Including Features of the Electronics Involved

In the present case, the linearity of the ammonium power spectra over the whole range of available scales (Figure 3) shows the absence of any kind of noise contaminations by the electronics or the processing chain, in which case the high-frequency part of the spectra would have shown a roll-off towards the noise level of the involved electronics [44]. The absence of this characteristic signature of high-frequency noise demonstrates that our sampling frequency is well above the electronic noise level. On the other hand, both low- and high-density patches perceptible from Figure 1 include more than one point (i.e., from 5 to 20), and this excludes the occurrence of electronic spikiness. These patches correspond to 5 to 20 points, then to time scales bounded between 15 and 60 s, and considering the mean flows experienced during our sampling experiments, to spatial scales bounded between 0.2 and 60 m.

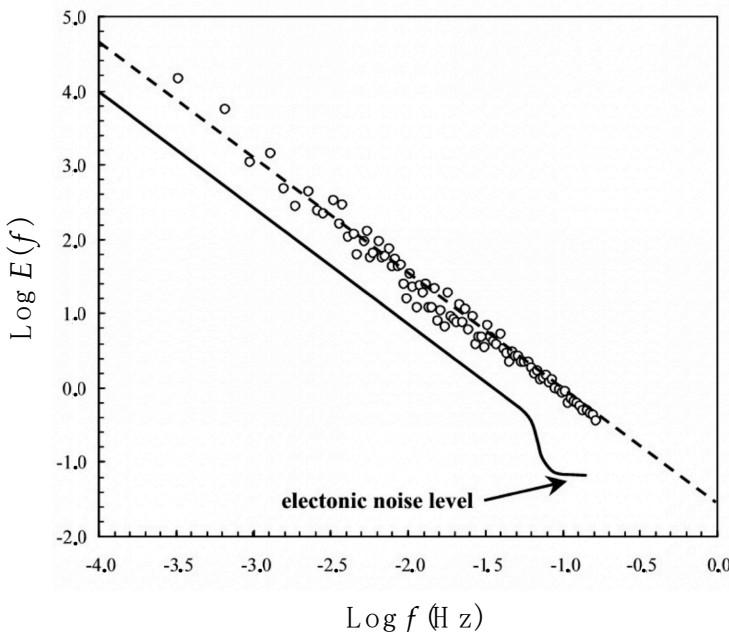

**Figure 3.** The power spectrum $E(f)$, where f is frequency, of an ammonium time series recorded in 1997. The strong linearity of the power spectrum indicates a scaling behavior over the whole range of scales. The spectrum expected in case of noise contamination by the electronics of the processing chain, presenting a high-frequency roll-off towards the electronic noise level, is shown for comparison.

### 2.2.3. The Turbulent and/or Molecular Diffusion Occurring in the Plastic Tubing of the Pumping Apparatus

First, we investigated the importance of the forces due to viscosity by calculating the Reynolds number Re, Re = $ud/v$, where $u$ is the flow speed in the plastic tube ($u = 12.5$ mm s$^{-1}$), $d$ the diameter of the tube ($d = 1.5$ mm), and $v$ is the kinematic viscosity ($v = 10^{-6}$ m$^2$ s$^{-1}$). The low Reynolds number associated with our pumping apparatus (Re = 18.75) indicates that no turbulent mixing occurs during the pumping process in the plastic tubing. We subsequently assessed the diffusion length scale, $L$, travelled by $NH_4^+$ molecules in the plastic tubing during the pumping process due to the molecular diffusion; $L$ is defined as $L = (Dt)^{0.5}$, where $D$ is the molecular diffusion ($D = 10^{-9}$ m$^2$ s$^{-1}$) and $t$ the diffusion time scale (i.e., the time needed to bring seawater through the Technicon Autoanalyzer II; i.e., $t = 20, 22$, and 16 min for the sampling experiments conducted in 1996, 1997, and 1998, respectively). The diffusion length scale $L$ is about $10^{-3}$ m (i.e., $L = 1.10 \times 10^{-3}$ m in 1996, $L = 1.15 \times 10^{-3}$ m in 1997, and $L = 0.98 \times 10^{-3}$ m in 1998). This length scale is typically five orders of magnitude lower than the inter-sample bubbling used by the Technicon Autoanalyzer II [28]. As a consequence, it is stressed that the low Reynolds number and minute diffusion length scale characteristic of our pumping procedure cannot induce any bias related to turbulent diffusion nor molecular diffusion.

### 2.2.4. The Mixing Induced by the Boundary Layer Occurring around the Hull of the Vessel

This potential bias has been investigated by estimating the thickness of the boundary layer generated by the tidal current flowing around the hull of the ship, which have been compared to the 1-m distance chosen for the seawater intake. The thickness of a boundary layer, $\delta$, increases with increasing distance from the ship bow according to $\delta = (xv/v)^{1/2}$ [45], where $x$ is the distance from the ship bow where water has been continuously taken ($x = 15$ m), $v$ the kinematic viscosity and $v$ (m s$^{-1}$) the tidal current speed. We then estimated $\delta$ for the range of velocities (0.05–1.50 m s$^{-1}$) experienced during the seawater pumping experiments as being in the range 0.32–1.73 cm. The potential influence of such minute boundary layer thickness on the temporal patterns of the nitrite measurements taken 1 m away can be obviously neglected.

## 2.3. Stochastic Quantification of Intermittent Nutrient Distribution

### 2.3.1. Theoretical Analysis

The fluctuations of passive scalars such as temperature, salinity, and a priori phytoplankton cells occurring under the influence of fully developed homogeneous tri-dimenstional turbulence have widely been described in Fourier space using power spectral analysis as $E(f) \propto f^{-\beta}$ where f is the frequency ($s^{-1}$) and $\beta \approx 5/3$. A power spectrum fundamentally quantifies the amount of variability occurring in different frequency bands. When all or parts of the spectrum follow the abovementioned power law, or more generally a power law of the form $E(f) \propto f^{-\beta}$ where $\beta$ can diverge from the theoretical expectation $\beta \approx 5/3$, this indicates the absence of any characteristic time scale in the range of scales where the power law applies.

Spectral analysis is, however, limited to a second-order statistic, hence characterizes very poorly intermittent fluctuations (i.e., occasional and unpredictable large peaks separated by very low values; see Figure 1) that are fundamentally non-Gaussian [4]. Spectral analysis can be generalized in real space using the $q$th order structure functions that have been extensively described and illustrated elsewhere [4]. For a concentration of nutrient C fluctuating in time, the $q$th order structure functions are defined as $\langle (\Delta C(\tau))^q \rangle \propto \langle |C(t+\tau) - C(\tau)|^q \rangle$, where the quantity $\langle (\Delta C(\tau))^q \rangle$ is the $q$th order statistical moments of the fluctuations of the quantity C at the time scale $\tau$.

The structure function exponents $\zeta(q)$ are defined as $\langle (\Delta C(\tau))^q \rangle \propto t^{\zeta(q)}$, and the scaling exponents $\zeta(q)$ are given by the slope of the linear trends of $\langle (\Delta C(\tau))^q \rangle$ versus $\tau$ in a log–log plot. Specifically, the exponent $\zeta(1) = H$ defines the scale-dependency of the average fluctuations, that is if $H \neq 0$, the fluctuations will depend on the time scale. In turn, the power spectral slope $\beta$ is linked to the second moment $\zeta(2)$ as $\beta = 1 + \zeta(2)$. The function $\zeta(q)$ is linear for monoscaling (i.e., monofractal) processes such as Brownian motion (($\zeta(q) = q/2$) and non-intermittent turbulence ($\zeta(q) = q/3$). In contrast, $\zeta(q)$ is non-linear and convex for multiscaling (i.e., multifractal) processes [4]. The convexity of the function $\zeta(q)$, i.e., $\zeta(q) = qH - K(q)$, corresponds to the intermittent (i.e., patchy) deviation from homogeneity, in which case $\zeta(q) = qH$. The parameter $K(2)$, often referred to as $\mu$, is called the *intermittency parameter* and is typically in the range 0.1–0.3 for passive scalars in turbulent flows [4,28,46–48].

Practically, a function $\zeta(q)$ diverging from linearity characterizes a heterogeneous distribution with a few dense patches over a wide range of low-density patches. As such the function $\zeta(q)$ is used in this study as an index of nutrient patchiness: the more convex $\zeta(q)$ is, the more patchy or intermittent the nutrient distribution is.

### 2.3.2. Intermittent Ammonium Distribution vs. Turbulence Intensity

All the time series investigated significantly diverged from a non-intermittent distribution (Figure 4). Their stochastic properties were related to turbulent mixing intensities, with a clear increase in nonlinearity (i.e., an indication of more intermittent distributions) under conditions of decreasing turbulence. The intermittency parameters $K(2)$ consequently significantly differed according to the velocity of the tidal flow v, with $K(2) = 0.21 \pm 0.01$, $K(2) = 0.14 \pm 0.01$, and $K(2) = 0.06 \pm 0.01$ for $v = 0.20$, 0.5, and 1 m $s^{-1}$, respectively.

These observations confirmed previous observations conducted on nitrite distribution [28] and further showed that (i) $NH_4^+$ was consistently heterogeneously distributed for turbulence intensities typically ranging from $10^{-7}$ to $10^{-4}$ $m^2$ $s^{-3}$, and (ii) for similar concentrations $NH_4^+$ distributions were significantly more heterogeneous for turbulence intensities ranging from $2.1 \times 10^{-7}$ $m^2$ $s^{-3}$ to $7.2 \times 10^{-6}$ $m^2$ $s^{-3}$ than for turbulence intensities ranging from $4.8 \times 10^{-5}$ $m^2$ $s^{-3}$ to $3.4 \times 10^{-4}$ $m^2$ $s^{-3}$.

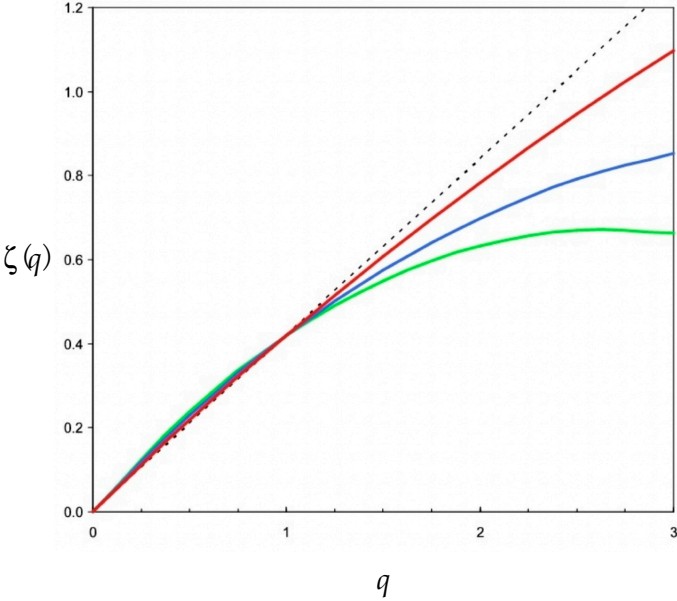

$$q$$

**Figure 4.** Illustration of the nonlinearity of the empirical function $\zeta(q)$ estimated for ammonium time series under different conditions of flows, i.e., 1 m s$^{-1}$ ($\varepsilon \sim 6 \times 10^{-4}$ m$^2$ s$^{-3}$, red), 0.5 m s$^{-1}$ ($\varepsilon \sim 7.5 \times 10^{-5}$ m$^2$ s$^{-3}$, blue), and 0.2 m s$^{-1}$ ($\varepsilon \sim 4.8 \times 10^{-6}$ m$^2$ s$^{-3}$, green) in comparison to the theoretical linear non-intermittent case $\zeta(q) = qH$ (black dotted line). Note that the empirical function $\zeta(q)$ are consistently non-linear and convex, with an increasing divergence to the theoretical linear case $\zeta(q) = qH$, with increasing values of the statistical order of moment $q$. The dissipation rate of turbulent energy $\varepsilon$ (m$^2$ s$^{-3}$) were estimated from the velocity of the tidal flow $v$ as $\varepsilon = \phi v^3/z$, where $z$ is the depth of the water column ($z \sim 10$ m) and $\phi$ is a constant, $\phi = 0.006$; see Equation (15) below.

## 3. Nutrient Patches and Phytoplankton Uptake

### 3.1. Phytoplankton Nutrient Uptake in a Steady-State Environment

The most common functional formulation chosen to describe nutrient uptake rate ($J$) is the rectangular hyperbola [49]:

$$J = \frac{aC}{b + C} \tag{1}$$

where $C$ is the average extracellular nutrient concentration (mol L$^{-1}$), $a$ the maximum uptake rate (mol cell$^{-1}$ s$^{-1}$), and $b$ the extracellular nutrient concentration at half the maximum uptake rate (mol L$^{-1}$). Equation (1) is strictly equivalent to the Mickaelis-Menten equations used to describe enzyme kinetics mechanistically as:

$$J = \frac{V_{\max}C}{K_m + C} \tag{2}$$

where $V_{\max}$ is the maximum uptake rate (mol cell$^{-1}$ s$^{-1}$) and $K_m$ the extracellular nutrient concentration at half the maximum uptake rate (mol L$^{-1}$). A variety of formulations have further been proposed to take into account the effect of extracellular physical processes, such as fluid motion, molecular diffusivity of nutrient and phytoplankton shape [50,51], and intracellular conditions, such as internal nutrient quotas and rates of biochemical reactions [52], on nutrient uptake. This ultimately led to reformulate nutrient uptake as [53]:

$$J_{i,j} = \psi_j D_i Sh_{i,j} C_i \left( \frac{Q_{i,j}^{max} - Q_{i,j}}{Q_{i,j}^{max}} \right) = \frac{dQ_{i,j}}{dt} \tag{3}$$

where $J_{i,j}$ is the uptake rate of nutrient $i$ to phytoplankton species $j$ ($\mu$mol cell$^{-1}$ s$^{-1}$), $\psi_j$ the diffusion shape factor for phytoplankton species $j$ (m), $D_i$ the molecular diffusivity of chemical species $i$ (m$^2$ s$^{-1}$), $Sh_{i,j}$ the Sherwood number (a non-dimensional measure of the passive flux of nutrient $i$ to phytoplankton species $j$ due to local fluid motion), $C_i$ the average extracellular concentration of chemical species $i$ ($\mu$mol L$^{-1}$), $Q_{i,j}^{max}$ the maximum internal cell quota of nutrient $i$ in phytoplankton species $j$ ($\mu$mol cell$^{-1}$), and $Q_{i,j}$ is the internal cell quota of nutrient $i$ in phytoplankton species $j$ ($\mu$mol cell$^{-1}$).

As stressed by Currie [39], and to the best of our knowledge, no experimentally validated model of nutrient uptake under fluctuating nutrient conditions exists. Moreover, Michaelis-Menten kinetics do not satisfactorily explain uptake when nutrient concentrations fluctuate in time. Theoretical works suggested that nutrient patchiness will negatively impact uptake rates as uptake efficiency is lower at high nutrient concentrations than at low ones [39]. This is acceptable under the general assumption that the parameters of the Michaelis-Menten kinetics remain constant irrespective of ambient nutrient concentration [39]. This assumption is, however, unrealistic given the known abilities of nutritionally limited phytoplankton cells to enhance their uptake of nutrients in the presence of ephemeral point source [40–42]. We thus introduce hereafter a novel model that may account for the observed surge-uptake of nutrients based (i) on the detailed stochastic properties of intermittent nutrient distributions, and (ii) on a simple adaptive representation of phytoplankton surge-uptake for nutrients.

### 3.2. A Simplified Model of Nutrient Surge Uptake in an Intermittent Environment

#### 3.2.1. Theoretical Formulation of the Stochastic Properties of Intermittent Nutrient Distribution

We only review here the main properties of intermittent (i.e., multifractal) fields. More details on the use of multifractal algorithms to marine ecology studies and what can be concluded from their use can be found elsewhere [4,54]. A main property of an intermittent field is that its fluctuations are not destroyed by smoothing at any scale, until the outer scale of the system is reached. For a given nutrient concentration $C$, this means that the intermittent field $C$ (see Figure 1) averaged over a scale $l$ will have a scale-dependent value denoted as $C_l$, or as $C_\lambda$. Here we introduce a non-dimensional scale ratio $\lambda$ ($\lambda = L/l$), which is the ratio between an external length scale $L$ and a targeted length scale $l$ within the inertial sub-range, i.e., $l_k \leq l \leq L$ where the Kolmogorov length scale is expresses as $l_k$; see [4,54] for further details. We assume in this analysis $\lambda \gg 1$. The scale-dependent multifractal field $C_\lambda$ can be described by its probability distribution, or equivalently, by its statistical moments $\langle (C_\lambda)^q \rangle$, where we consider any $q \geq 0$. These moments can be scaled with the scale ratio $\lambda$, as [4,54]:

$$\langle (C_\lambda)^q \rangle = C_0^q \lambda^{K(q)}, \tag{4}$$

Considering a continuous range of values of $q \geq 0$, Equation (4) is valid only for scales belonging to the inertial subrange, thus for $1 \leq \lambda \leq \Lambda$, where $\Lambda = L/l_B$ is the maximum scale ratio, between the larger outer scale $L$ and the Batchelor scale $l_B$, i.e., the smallest length scales of fluctuations in scalar concentration (nutrient concentration in the present case) that can exist before being dominated by molecular diffusion. The angle brackets '$\langle . \rangle$' in Equation (4) indicate statistical averaging, $C_0^q = \langle C_\lambda \rangle$ is the mean of the multifractal process $C_\lambda$, and $K(q)$ is a concave scale-invariant moment function that satisfies $K(0) = 0$ and $K(1) = 0$ [4]. The function $K(q)$ describes the whole statistics of the process, in an equivalent manner as the probability distribution. As stressed above, the second moment $K(2)$ is usually used as an intermittency parameter and referred to as $\mu$. Subsequently, Equation (4) can be used to evaluate the average of any polynomial function $f(C_\lambda)$ of the multifractal field $C_\lambda$ as [54]:

$$f(C_\lambda) = \sum_{p=1}^{N} a_p (C_\lambda)^p \tag{5}$$

where $a_p$ are constants, and $p$ the polynomial order of the function $f(C_\lambda)$ bounded between 1 and $N$. Averaging the function $f(C_\lambda)$ finally leads to:

$$\langle f(C_\lambda) \rangle = \sum_{p=1}^{N} a_p C_0^p \lambda^{K(p)} \tag{6}$$

### 3.2.2. A Simplified Model for Nutrient Surge Uptake under Intermittent Conditions

Under the assumption of statistical independence between nutrient, phytoplankton, and turbulent fields [28,54], the uptake of nutrient can be thought as a two-step process: (i) the encounter between a phytoplankton cell and nutrient molecules and (ii) the actual nutrient uptake. By analogy with predator-prey encounter theory, the encounter rate E between a phytoplankton cell and nutrient molecules is expressed as:

$$E = \beta C_\lambda \tag{7}$$

where $\beta$ is the encounter kernel due to turbulence and behavior and $C_\lambda$ the ambient intermittent nutrient concentration. Now, once a phytoplankton-nutrient encounter occurred, the ability of phytoplankton cells to enhance their uptake of nutrients in the presence of ephemeral point source was expressed following the general mechanistic formulation of Baird and Emsley [53] as:

$$J \propto E \times C_\lambda \tag{8}$$

where $J$ is the instantaneous nutrient uptake rate ($\mu$mol cell$^{-1}$ s$^{-1}$). Equations (7) and (8) subsequently simply rewrite as:

$$J \propto C_\lambda^2 \tag{9}$$

The average uptake rate of a phytoplankton cell exposed to an intermittent nutrient distribution will further be expressed as:

$$\langle J \rangle_{\text{inter}} \propto C_0^2 \lambda^{K(2)} \tag{10}$$

where $C_0 = \langle C_\lambda \rangle$ is the average nutrient concentration experienced by phytoplankton cells. In contrast, the average uptake rate of a phytoplankton cell exposed to a homogenous nutrient distribution is given by:

$$\langle J \rangle_{\text{homo}} \propto C_0^2 \tag{11}$$

It directly comes from the comparisons of Equations (10) and (11) that:

$$\langle J \rangle_{\text{inter}} > \langle J \rangle_{\text{homo}} \tag{12}$$

### 3.2.3. Surge Uptake under Homogenous and Intermittent Nutrient Distribution: The Turbulent History Hypothesis

We estimated the potential effect of intermittent nutrient distributions on phytoplankton uptake from Equations (10) and (11) as:

$$\frac{\langle J \rangle_{\text{inter}}}{\langle J \rangle_{\text{homo}}} \propto \lambda^{K(2)}, \tag{13}$$

using the values of the intermittency parameter $K(2)$ and the scale-ratio $\lambda$ estimated for ammonium distributions under varying conditions of flow velocities. This resulted in increases in the uptake rates by 4.2-fold, 2.65-fold, and 1.48-fold for flow velocities of 0.2, 0.5, and 1 m s$^{-1}$, respectively. Assuming nutrient limitation, and in the absence of significant differences in the average ammonium concentrations (i.e., $C_0 = \langle C_\lambda \rangle$), these results imply that under elevated turbulent conditions, phytoplankton cells would experience a low-density background of ammonium concentration, resulting in low uptake rates. These cells are then more likely to be nutrient depleted, and would hence exhibit low affinity for NH$_4^+$ [41,50,55,56]. An optimal uptake strategy would thus consist in an increase in transport

rate toward the cell [56,57] through the activation of nitrogen-regulated proteins [58,59]. In contrast, under low turbulent conditions, phytoplankton cells would experience high density nitrogen patches over a wide range of impoverished water, become transport limited in their uptake rates [50,55–57], and develop a higher affinity for $NH_4^+$. Note that affinity is here employed *sensu* Healey (1980), i.e., low affinity and high affinity respectively relate to high and low values of $K_m$ in Equation (2) [60].

The results of our observations and theoretical model suggest that:

- For the same concentrations, the distribution of ammonium is controlled by turbulence, switching from a more homogeneous to a more heterogeneous distribution respectively under high and low turbulence intensities. This is consistent with previous observations conducted on nitrite and phytoplankton concentrations [11,28].
- The turbulent regime experienced by phytoplankton cells, here referred to as their 'turbulent history' will condition their affinity to ammonium and its transport rate.
- As a consequence, any uptake experiments conducted on natural phytoplankton communities would be intrinsically influenced, if not biased, by their turbulent history. In order to validate our mechanistic hypotheses, we specifically designed a field experiment devoted to assess the surge uptake rates of natural phytoplankton communities under ammonium limitations when exposed to ammonium pulses of low and high concentrations.

## 4. Empirical Validation: A Case Study from a Turbulent Coastal Sea, the Eastern English Channel

### 4.1. Field Site and Sampling Strategy

The eastern English Channel (EEC) is a tidally-mixed coastal ecosystem where strong tidal currents and shallow waters lead to turbulent kinetic energy dissipation rates varying between $10^{-6}$ and $10^{-4}$ m$^2$ s$^{-3}$ [11,28]. This area is also structured along a north/south gradient (Figure 5); the northern Strait of Dover being more turbulent than the sheltered waters of the southern Bay of Somme [61].

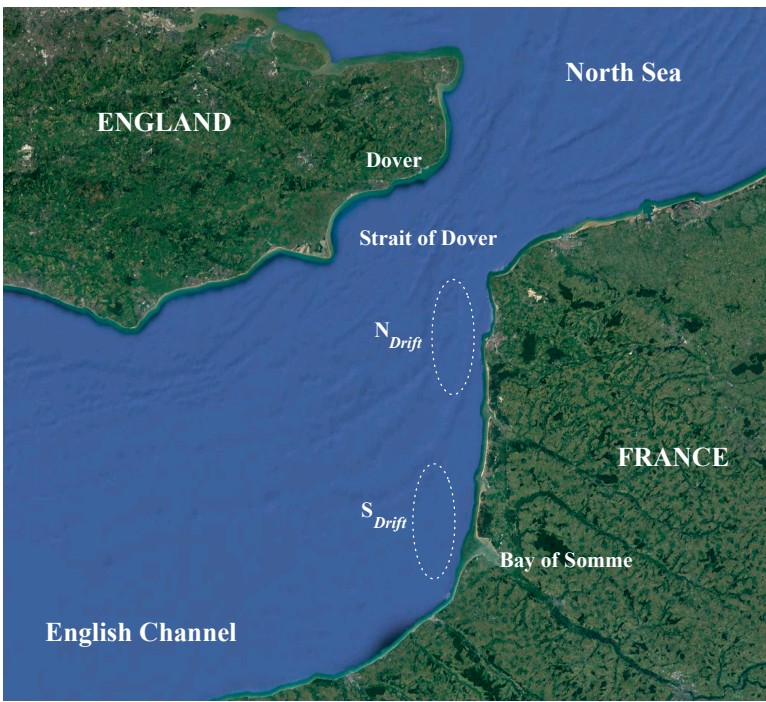

**Figure 5.** Field site and location of the drift area in the northern ($N_{drift}$) and southern ($S_{drift}$) part of the eastern English Channel.

Three multidisciplinary cruises were conducted during the late phase of the 2003 spring phytoplankton bloom (i.e., April, May, and July; Table 1) adrift aboard the N/O "Côtes de la Manche"

(CNRS, INSU) in the Strait of Dover and the Bay of Somme (Table 1, Figure 5). Hereafter, sampling sites are referred to as N and S, where "N" and "S" are the northern (Strait of Dover) and southern (Bay of Somme) water masses of the eastern English Channel. ARGOS buoy equipped with a GPS system was followed during 24 h for each period and location. Temperature (°C) and salinity profiles were acquired at each sampling station with a SBE 25 Sealogger CTD (Sea-Bird Scientific, Bellevue, WA, USA). Water samples were taken from sub-surface (1 m) using 30-L Niskin bottles. No sampling was conducted in the northern station in April (Table 1).

**Table 1.** In situ conditions during North (N) and South (S) drift cruises; dates, locations (latitude/longitude), depths and tidal conditions (ST: spring tide; NT: neap-tide; F: flood tide and E: ebb tide), vertically averaged (±SE) salinity (S) and temperature (T), nitrite + nitrate ($NO_2^- + NO_3^-$), and ammonium ($NH_4^+$) concentrations (µM), and chlorophyll *a* ([Chl *a*]) surface concentrations (µg L$^{-1}$), on sampling stations. (-) No data was available for N in April. DL: Detection Limit.

| Area | Date | Latitude | Longitude | Depth (m) | Tide | F/E | S(PSU) | T (°C) | Nitrite + Nitrate (mM) | Ammonium (mM) | Chlorophyll *a* (mg L$^{-1}$) |
|---|---|---|---|---|---|---|---|---|---|---|---|
| S | April 24 | 50°37′688 N | 1°25′963 E | 17.5 | NT | E | 33.9 (0.3) | 9.9 (0.2) | 0.54 | 0.2 | 7.22 |
| N | May 11 | 50°41′052 N | 1°26′988 E | 33 | NT | E | 34.7 (0.0) | 11.1 (0.1) | <DL | 0.4 | 1.7 |
| S | May 14 | 50°20′489 N | 1°24′633 E | 19.5 | ST | E | 34.1 (0.0) | 11.8 (0.0) | 0.11 | 0.78 | 6.11 |
| N | July 7 | 50°50′240 N | 1°28′192 E | 52 | NT | E | 33.3 (0.2) | 18.7 (0.2) | <DL | 1 | 3.94 |
| S | July 9 | 50°18′432 N | 1°22′336 E | 15.7 | NT | E | 34.2 (0.1) | 17.6 (0.1) | 0.1 | 0.72 | 5.11 |

### 4.2. Chemical and Biological Environment

For dissolved inorganic nitrogen ($NO_2^- + NO_3^-$), 10-mL water samples were frozen at −20 °C immediately after collection, and analyzed in the laboratory with an auto-analyzer (Alliance Integral Futura, AMS Alliance, Frépillon, France). Ammonium ($NH_4^+$) concentrations were determined manually on 100-mL water samples [62]. Water samples (200 mL) were filtered through glass-fiber filters (Whatman GF/C, GE Healthcare Life Sciences, Little Chalfront, UK), immediately frozen (−20 °C), and chlorophyllous pigments were subsequently extracted with 90% acetone (5 mL) in the dark at 4 °C, assayed in a spectrophotometer (UVIKON 940, Kontron instruments®, Montigny-le-Bretonneux, Paris, France) and chlorophyll a concentrations calculated following UNESCO standard calculation [63].

### 4.3. Quantifying Surge Uptake Rates

Ammonium ($NH_4^+$) was used to characterize surge uptake processes. This nitrogen source was specifically chosen since (i) phytoplankton cells are susceptible to be frequently exposed to pulses particularly under nitrogen limited conditions [64], (ii) various biological processes (i.e., excretion by auto/heterotrophic plankton, bacterial activities) are likely to release $NH_4^+$ in the water column and (iii) $NH_4^+$ is the most limiting nitrogen source in the eastern English Channel, see e.g., [65,66]. Sub-surface water was sampled using a 15-L Niskin bottle, and the surge uptake experiment conducted in nine 1-L bottles filled with in situ seawater; three were used as a control (i.e., without enrichment), three received a low ammonium (as $(NH_4)_2SO_4$) concentration pulse (0.5 µM), and the remaining three with a high ammonium concentration pulse (2 µM). All these bottles were subsequently incubated on the boat deck under natural light and temperature. An additional liter was used to assess the standing stock of auto-, hetero-, and mixotrophic protists.

Ammonium concentrations, [$NH_4^+$], were measured manually [58] on 10-mL sub-samples taken in each incubation bottle before $NH_4^+$ addition ($t_0$) and 5 min after the pulse ($t_5$). Short-term incubations were chosen to observe surge uptake processes, known to take place within a few minutes after

a pulse [36,41]. Surge uptake rates $\rho_{surge}$ ($\mu$mol(NH$_4^+$) $\mu$gC$^{-1}$ min$^{-1}$) were estimated from NH$_4^+$ consumption, and normalized by phytoplankton biomass as:

$$\rho_{surge} = \left[\left(\left[\text{NH}_4^+\right]_{t_5} - \left[\text{NH}_4^+\right]_{t_0}\right)/\Delta t\right]/C \qquad (14)$$

where $\left[\text{NH}_4^+\right]_{t_0}$ and $\left[\text{NH}_4^+\right]_{t_5}$ are NH$_4^+$ concentrations ($\mu$M) at $t_0$ and $t_5$, $\Delta t$ the time of incubation ($\Delta t = 5$ min), and C the phytoplankton biomass ($\mu$gC L$^{-1}$) estimated at each sampling site. Phytoplankton biomass C was estimated from the abundance of auto- and hetero/mixotrophic protists. Cells were measured with an eyepiece micrometer and corresponding biovolumes were calculated by relating the shape of organisms to a standard geometric form. Biovolumes were converted to carbon biomass following [67,68].

### 4.4. Quantifying the Turbulent History of Phytoplankton Cells

The dissipation rate of turbulent energy induced by the tidal flow $\varepsilon$ (m$^2$ s$^{-3}$) was estimated as [69]:

$$\epsilon = \phi u^3 / z \qquad (15)$$

where $\phi$ represent the fraction of the tidal energy used for vertical mixing ($\phi = 0.006$, [70]), u the drift speed between two successive stations, which corresponds to M2 depth-averaged tidal velocity (m s$^{-1}$) and $z$ the water column depth (m).

### 4.5. Turbulent History, Nutrient Patchiness and Phytoplankton Uptake Rates

Salinity showed a stationary behavior fluctuating between 33.3 (N-July) and 34.7 (N-May) over the survey. In contrast, temperature exhibited a clear seasonal cycle, increasing gradually from $9.9 \pm 0.2$ °C (S-April) to $18.7 \pm 0.2$ °C (N-July; Table 1). These temperature and salinity values are consistent with previous measurements done at the seasonal scale in the EEC [65,71–74].

Chlorophyll *a* concentrations ranged between 1.70 $\mu$g L$^{-1}$ (N-May) and 7.22 $\mu$g L$^{-1}$ (S-April; Table 1). These relatively low Chl *a* concentrations are typical of the conditions encountered during the late phase of the spring bloom in the EEC [72,73]. The composition of phytoplankton populations was homogeneous and dominated by the large diatoms *Guinardia striata* and *Rhizosolenia imbricata* over the duration of the survey [61]. In accordance with previous studies conducted in the EEC [61,62], nitrogen concentrations (NO$_2^-$ + NO$_3^-$ and NH$_4^+$) remained low (i.e., typically $\leq 1$ $\mu$M) throughout the survey (Table 1), suggesting that phytoplankton communities were potentially nitrogen-limited.

Turbulent energy dissipation rates $\varepsilon$ were highly variable over the survey, ranging between $1.26 \times 10^{-6}$ m$^2$ s$^{-3}$ (S-April) and $8.59 \times 10^{-5}$ m$^2$ s$^{-3}$ (N-July; Figure 6). These values are in the range of turbulence intensities reported in this area, i.e., $10^{-7} < \varepsilon < 10^{-4}$ m$^2$ s$^{-3}$ [11,48]. Based on integrated turbulence intensities experienced by phytoplankton cells during the 5–6 h preceding their sampling, we discriminated two groups of stations: (i) N-July, S-May, and S-July characterized by high turbulent levels (i.e., $\varepsilon > 10^{-5}$ m$^2$ s$^{-3}$) and (ii) N-May and S-April characterized by lower turbulence intensities (i.e., $\varepsilon < 10^{-5}$ m$^2$ s$^{-3}$; Figure 6). This critical turbulence intensity has been chosen as it has previously been identified as the turbulence threshold above and below which the level of patchiness of nitrite distributions were significantly different [28]. Specifically, these early observations conducted on nitrite time series were confirmed and specified by the ammonium distributions continuously sampled in 1996, 1997, and 1998. These distributions were consistently patchy irrespective of turbulence intensity (see Figures 1 and 4), and for similar (NH$_4^+$) concentrations, were significantly more heterogeneous for turbulence intensities ranging from $2.1 \times 10^{-7}$ m$^2$ s$^{-3}$ to $7.2 \times 10^{-6}$ m$^2$ s$^{-3}$ than for turbulence intensities ranging from $4.8 \times 10^{-5}$ m$^2$ s$^{-3}$ to $3.4 \times 10^{-4}$ m$^2$ s$^{-3}$.

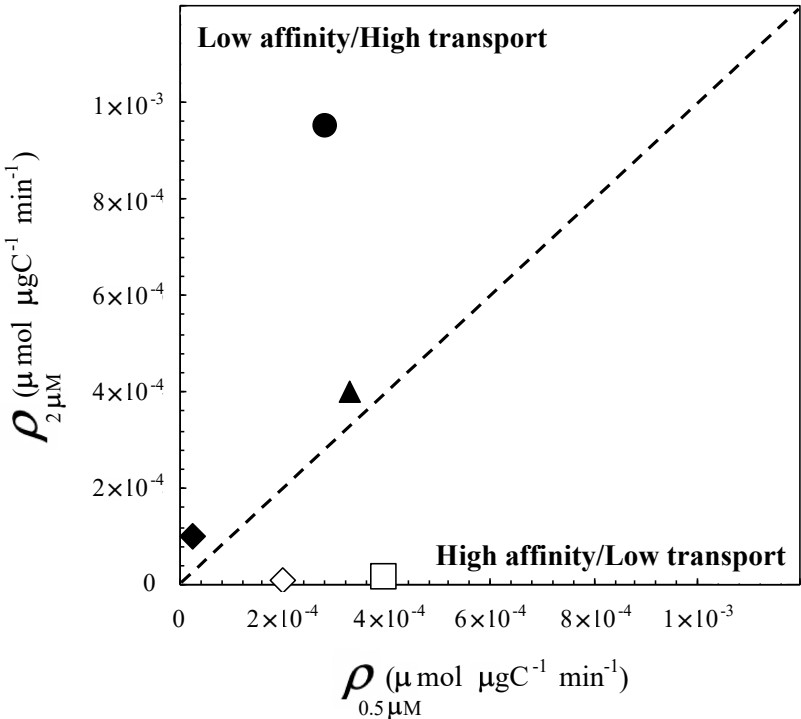

**Figure 6.** Surge uptake rates after a pulse of 2 μM ($\rho_{2\mu M}$; μmol μgC$^{-1}$ min$^{-1}$) vs. surge uptake rates after a pulse of 0.5 μM ($\rho_{0.5\mu M}$; μmol μgC$^{-1}$ min$^{-1}$). The closed symbol are the surge uptake rates measured at stations characterized by high turbulence levels $\varepsilon$ (i.e., $\varepsilon > 10^{-5}$ m$^2$ s$^{-3}$); S-May (diamond), S-July (triangle) and N-July (circle). The open symbols are the surge uptake rates measured at stations characterized by low turbulence levels (i.e., $\varepsilon < 10^{-5}$ m$^2$ s$^{-3}$); N-May (diamond) and S-April (square). The size of the symbol represents the 99% confidence intervals of triplicated surge uptake rates.

The control experiment run without ($NH_4^+$) enrichment did not exhibit any surge uptake, i.e., $\rho_{surge} = 0$; see Equation (14). Surge uptake rates measured following $NH_4^+$ enrichment ranged from 0 to $9.54 \times 10^{-4}$ μmol($NH_4^+$) μgC$^{-1}$ min$^{-1}$ and appeared to be strongly related to the interplay between $NH_4^+$ concentrations and turbulence intensities (Figure 6). Cells exposed to high turbulent intensities exhibited maximum surge uptake rates after a pulse of 2 μM (Figure 6), suggesting a low affinity for $NH_4^+$ and a subsequent high transport rate. In contrast, cells exposed to lower turbulence intensities before the uptake experiment showed higher surge uptake rates after a pulse of 0.5 μM (Figure 6). This suggests a high affinity for $NH_4^+$ and a low transport rate of $NH_4^+$. Noticeably, no surge uptake was observed following a 2 μM $NH_4^+$ enrichment under the two lowest conditions of turbulence history (Figure 6). While the resolution of this specific issue is beyond the scope of this preliminary study, this observation suggests that turbulence history may play a more critical role than the actual density of the nutrient patches encountered by phytoplankton cells in conditioning their surge uptake abilities. This hypothesis is actually further supported by our observations that showed a consistently higher surge uptake under conditions of high (i.e., 2 μM) $NH_4^+$ enrichment and low turbulent history than under conditions of low (i.e., 0.5 μM) $NH_4^+$ enrichment and high turbulence history.

An undisputed effect of microscale turbulence on phytoplankton is the shear-controlled increase in the passive nutrient fluxes towards phytoplankton cells with turbulence intensities [51,74]. The surge uptake patterns observed here may, however, also rely on the interplay between turbulence and the spatial distribution of dissolved nitrogen. The distribution of dissolved inorganic nutrients has thus been shown to be controlled by the intensity of turbulent mixing, high and low nutrient patchiness were identified under low and high turbulent conditions, respectively [28]; see also Figures 1 and 4.

At scales relevant to individual cells, phytoplankton cells exposed to high and low turbulent intensities may then be adapted to more evenly and more patchy nutrient distributions, respectively.

Under nutrient limitation, phytoplankton cells exposed to high turbulence intensities would experience a low background of evenly distributed nutrients and would thus exhibit low affinity for $NH_4^+$ [41,50,55,56]. An optimal uptake strategy would thus consist in an increase in transport rate toward the cell [56,57] through the activation of nitrogen regulated proteins, e.g., [58,59]. To date, there is no evidence of the role of microscale shear on the activation of nitrogen regulated proteins, and it is therefore highly speculative, although tempting, to suggest a connection between the elevated shear $\gamma$ ($\gamma = (\varepsilon/\nu)^{0.5}$, where $\varepsilon$ is the dissipation rate of turbulent energy and $\nu$ the kinematic viscosity, ca. $10^{-6}$ m$^2$ s$^{-1}$) experienced by phytoplankton cells prior to sampling ($\gamma > 3.2$ s$^{-1}$, i.e., when $\varepsilon > 10^{-5}$ m$^2$ s$^{-3}$) and an increase in transport rates. In contrast, cells exposed to low turbulent intensities would experience high density nitrogen patches over a wide range of impoverished water, become transport limited in their uptake rates [41,50,55,56] and develop a higher affinity for $NH_4^+$, an assumption consistent with our observations (Figure 6).

## 5. Conclusions

These results indicate that turbulence controls the microscale distribution of ammonium, switching from a more homogeneous to a more heterogeneous distribution respectively under high and low turbulence intensities for the same bulk concentration. In addition, using a new theoretical framework, we provide evidence of a potential interaction between small-scale turbulence, nutrient patchiness, and nutrient uptake by phytoplankton in natural waters. These preliminary results support the hypothesis that phytoplankton cells exposed to different turbulence levels would exhibit different abilities to use ephemeral nitrogen patches particularly under nitrogen limitation. Turbulence history is thus suggested as a potential fundamental lynch-pin in the control of the nutritive status of phytoplankton cells and as a consequence, any uptake experiments conducted on natural phytoplankton communities would be intrinsically influenced by their turbulent history. More fundamentally, these results highlight the importance of ocean variability at minute spatial and temporal scales in the structure and function of marine ecosystems. In this context, the approach presented here is consistent with previous studies showing how the understanding and subsequent modelling of intermittent distributions—or more generally small- and micro-scale variability—can enhance trophic transfer, interspecific competition, and eventually sustain biodiversity in plankton ecosystems [74–79].

It is finally stressed that simultaneous measurements of small-scale nutrient distributions (see e.g., [28]) and surge uptake rates from different environments, in particular oligotrophic ones, are needed to generalize our observations. The key role played by phytoplankton nutritive status in phytoplankton succession and species success (see e.g., [80]) nevertheless suggests that investigations of the effect of turbulent history on phytoplankton uptake rates may comprise areas of important future research.

**Author Contributions:** Conceptualization, M.S. and L.S.; methodology, M.S. and L.S.; laboratory analysis, M.S.; theoretical model development, L.S.; data interpretation, M.S. and L.S.; writing—original draft preparation, M.S.; writing—final draft preparation, review, and editing, M.S. and L.S.; funding acquisition, L.S. All authors have read and agreed to the published version of the manuscript.

**Funding:** This research was funded by the CPER "Phaeocystis", the PNEC "Chantier Manche Orientale-Sud Mer du Nord", and further supported under Australian Research Council's Discovery Projects funding scheme (project numbers DP0664681 and DP0988554). Professor Seuront is the recipient of an Australian Professorial Fellowship (project number DP0988554).

**Acknowledgments:** We acknowledge the captain and the crew of the NO "Côtes de la Manche" (CNRS-INSU) for their help during the sampling experiment. Two anonymous reviewers are acknowledged for their insightful comments and suggestions that greatly improved the quality of this work.

**Conflicts of Interest:** The authors declare no conflicts of interest.

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
