# Peer review of "Nutrient Patchiness, Phytoplankton Surge-Uptake, and Turbulent History: A Theoretical Approach and Its Experimental Validation"

_fluids, doi:10.3390/fluids5020080_

Round 1

Reviewer 1 Report

The authors report on experiments and a theoretical analysis of phytoplankton nutrient uptake, and its relation with turbulence conditions, nutrient patchiness, and concentrations. They specifically study the distribution and uptake of ammonium. They find that phytoplankton can regulate its uptake depending on the turbulence conditions.

I find this work interesting and worthy of publication. I recommend that the authors address the following points to increase readability.

- The authors should carefully define what is meant with nutrient affinity. Is it in the sense of [Healey, Microb, Ecol. 5, 281 (1980)] ? high uptake is consistent with high affinity.

- Line 162, a square root is missing in the expression for the boundary layer thickness.

- While it's clear that the function zeta(q) is convex, the plot in Fig. 4 is not. The authors should fix this incongruence.

- Lines 252-254, the definition of lambda and the statement L<l, conflict with the statement lambda>>1

- Can authors make contact with recent literature, such as:
1. Reigada, et al., Proc. R. Soc. Lond. B 270, 875 (2003).
2. Durham, et al. Nature Commun. 4, 1 (2013).
3. Breier, et al., Proc. Natl. Acad. Sci. USA 115, 12112 (2018).

that links turbulence with planktonic distributions?

Minor points:
- Eqs. 1-2, the comma can be mistaken for a prime on the variable C. Please add some space.

- Line 277, the subject "we" is missing.

Author Response

Dear Reviewer,

Please see the attached file for a detailed answer to your queries.

With kind regards,

L. Seuront & M. Schapira

Reviewer 2 Report

See attached .docx file.

Author Response

(The authors gave the same response as above.)

Round 2

Reviewer 2 Report

I thank the authors for their care in addressing my comments.  My substantial concerns regarding their field experimental work have been completely addressed, as it seems that the experimental design was perfectly sound and it was just that some important details were inadvertently omitted in the original text submission.  

I think that the manuscript is acceptable in its present form.

Author Response

As requested by the referee, we checked carefully the English and the spelling.

Kind regards,

L. Seuront & M. Schapira